# Comparison of different methods for monitoring changes in bone mineral density during follow-up measurements

Yen-Huai Lin[1,2], Michael Mu Huo Teng[1,2]*

**1** Department of Medical Imaging, Cheng Hsin General Hospital, Taipei, Taiwan, **2** School of Medicine, College of Medicine, National Yang Ming Chiao Tung University, Taipei, Taiwan

* mhteng@gmail.com

## Abstract

Monitoring bone mineral density (BMD) changes can be challenging when both the spine and hips are included in follow-up assessments. This study aimed to evaluate BMD changes using different monitoring approaches, including comparisons between the mean bilateral total hip BMD and total hip BMD of each hip separately, spinal BMD and mean bilateral total hip BMD, and spinal BMD and trabecular bone score (TBS). Bone density was measured using dual-energy X-ray absorptiometry, and bone quality was assessed using the TBS. A total of 1,105 postmenopausal women who underwent simultaneous follow-up BMD measurements of the spine and both hips were retrospectively analyzed. In follow-up BMD measurements, the discordance between the mean bilateral total hip BMD and the individual total hip BMDs (right and left hips) was 11.3% when the least significant change (LSC) of the mean bilateral total hip was applied. In contrast, this discordance increased to 19.5% when the average LSC of the right and left hips was used. Spine-hip discordance was also observed, with a 43.5% discordance rate between changes in spinal BMD and mean bilateral total hip BMD. Additionally, discordance between changes in spinal BMD and TBS was noted, with a discordance rate of 58.0%. Analyzing each hip separately is more sensitive to detecting changes than using the mean bilateral total hip BMD. When monitoring mean bilateral total hip BMD on repeat tests, the LSC of the mean bilateral total hip BMD is recommended. Discordance between spine and hip BMD, as well as between spinal BMD and TBS, was observed in follow-up measurements.

## Introduction

Bone mineral density (BMD), measured using dual-energy X-ray absorptiometry (DXA), plays an important role in diagnosing osteoporosis, identifying individuals at risk of fracture, assessing prognosis, and monitoring patients. DXA is the most validated and widely available tool for tracking changes in BMD. Follow-up BMD

**Data availability statement:** The dataset used in this study is owned by the Institutional Review Board (IRB) of Cheng Hsin General Hospital. While the IRB approved our use of the data for analysis, it did not grant permission for data sharing. As a result, we are unable to make the dataset publicly available. However, interested researchers may request access by contacting the IRB of Cheng Hsin General Hospital at chghirb@chgh.org.tw. Access will be provided under the same conditions as those granted to the authors, who do not have any special privileges.

**Funding:** This work was supported by a grant (CHGH115-N24) from the Cheng Hsin General Hospital, Taipei, Taiwan.

**Competing interests:** The authors have declared that no competing interests exist.

testing is commonly used to monitor patients at risk of fracture with or without treatment. However, practitioners often face uncertainty in monitoring BMD changes and interpreting findings when the spine and both hips are scanned on repeat tests. This uncertainty may lead to inappropriate follow-up of BMD measurements [1].

The International Society for Clinical Densitometry (ISCD) recommends that when both hips are scanned on repeat tests, the mean bilateral total hip BMD should be used for monitoring [2]. However, the differences between the monitored mean bilateral total hip BMD and the total hip BMD of each hip separately are uncertain. This uncertainty may lead to both overuse and underuse of DXA [3]. Additionally, the ISCD does not mention how to determine the least significant change (LSC) when monitoring the mean bilateral total hip BMD.

Various factors such as mechanical loading, pharmacological interventions, and degenerative changes can lead to site-specific variations in BMD over time, supporting the use of multiple-site assessments in clinical practice. The ISCD recommends measuring BMD at both the spine and hip for all patients [2]. However, the ISCD does not provide clear recommendations for interpreting discordant BMD changes when the spine and hips are assessed during follow-up. While such differences can sometimes be explained retrospectively, there is no standardized approach for incorporating them into clinical decision-making, especially when one site improves and another worsens. This ambiguity is particularly relevant in settings where treatment decisions rely on BMD trends.

The ISCD also recommends evaluating the bone quality using the trabecular bone score (TBS) [4]. When monitoring changes in the spinal BMD and TBS simultaneously on repeat tests, practitioners are often uncertain about the association between monitoring spinal BMD and TBS. Therefore, the aim of this study was to compare different monitoring methods (mean bilateral total hip BMD vs. total hip BMD of each hip separately, spinal BMD vs. mean bilateral total hip BMD, spinal BMD vs. TBS); additionally, we also aimed to determine LSC when monitoring mean bilateral total hip BMD in follow-up BMD measurements. Our aim was to highlight a gap in current guidelines, and our focus on this issue emphasizes the need for clearer guidance in interpreting longitudinal BMD data.

## Methods

### Participants

DXA images of Chinese participants who underwent scans at our hospital between 2019 and 2021 were retrospectively reviewed. In this study, we included postmenopausal women who had their spine and both hips scanned simultaneously and underwent follow-up BMD measurements. Participants who underwent follow-up BMD measurements within one year and had a body mass index (BMI) outside the 15–37 kg/m$^2$ range were excluded. Finally, 1,105 participants were enrolled in the study and the mean follow-up time was 23 months. This study was approved by the institutional review board of Cheng Hsin General Hospital (IRB no. (914)110A-60), and we accessed the data retrospectively on February 1st, 2022. The requirement of informed consent was waived due to the retrospective nature of this study.

### Bone mineral density

The BMD of the spine and both hips in all patients was measured using a DXA scanner (Horizon W; Hologic Inc., Bedford, MA, USA). We followed the official 2023 ISCD positions, and 30 healthy patients were recruited to perform a precision analysis with patient repositioning between scans [2]. The LSC of the spine, right total hip, left total hip, and mean bilateral total hip were 0.031 g/cm$^2$, 0.023 g/cm$^2$, 0.026 g/cm$^2$, and 0.019 g/cm$^2$ respectively, which were used to compare the differences between the baseline and follow-up BMD measurements.

### Trabecular bone score

The iNsight software (version 3.0.2.0; Medimaps, Geneva, Switzerland) was used to measure TBS, which utilized the spinal DXA images obtained using a DXA scanner [5]. We measured 30 patients twice, with patient repositioning after each scan to perform a precision analysis. The LSC of the TBS was 0.07, which was used to compare differences between baseline and follow-up TBS measurements.

### Statistical analysis

Cohen's kappa coefficient was used to quantify the level of agreement of BMD changes according to the different monitoring strategies in follow-up BMD measurements (spine BMD vs. mean bilateral total hip BMD). Additionally, Cohen's kappa coefficient was used to compare the agreement in changes between spinal BMD and TBS on repeat tests. The size of the Cohen's kappa coefficient was defined as poor, good, or excellent with scores of: $< 0.40$, 0.40–0.75, and $>0.75$ respectively. The statistical software SPSS for Windows (version 22.0; IBM Corp., Armonk, NY, USA) was used for all analyses.

## Results

The demographic characteristics of the participants were presented in Table 1. The mean age was 67.5 years. The mean spinal BMD, right total hip, and left total hip BMD were 0.793 g/cm$^2$, 0.725 g/cm$^2$, and 0.733 g/cm$^2$, respectively. The mean TBS score was 1.265.

Table 2 presented a comparison of the monitoring methods between the mean bilateral total hip BMD and the total hip BMD of each hip separately. The LSC of the right, left, and mean bilateral total hips were used to compare differences between baseline and follow-up BMD measurements. Among these participants, 10.3% had significantly greater BMD changes involving one side of the hip and significantly greater changes according to the mean bilateral total hip BMD. Of the participants, 9.6% had significantly lower BMD changes involving one side of the hip and significantly lower changes according to the mean bilateral total hip BMD. These finding were considered concordant with those of our study. In contrast, 5.5% of the participants had a significantly higher BMD change according to monitoring total hip BMD of each hip

**Table 1. Characteristics of the participants (n = 1,105).**

| | |
|---|---|
| Age (years, mean ± SD) | 67.5 ± 8.7 |
| Bone mineral density (g/cm$^2$, mean ± SD) | |
| Spine | 0.793 ± 0.14 |
| Right total hip | 0.725 ± 0.10 |
| Left total hip | 0.733 ± 0.11 |
| T-score (mean ± SD) | |
| Spine | −2.3 ± 1.3 |
| Right total hip | −1.8 ± 0.8 |
| Left total hip | −1.7 ± 0.9 |
| Trabecular bone score (mean ± SD) | 1.265 ± 0.09 |

**Table 2. Comparison of changes between monitoring mean bilateral total hip BMD and total hip BMD of each hip separately on repeat tests.**

| | | Changes of total hip BMD of each hip separately, n (%) | | | | | |
|---|---|---|---|---|---|---|---|
| | | (+, +) | (+, =) (=, +) | (=, =) | (-, =) (=, -) | (-, -) | (+, -) (-, +) |
| Changes of mean bilateral total hip BMD, n (%) | (+) | 166 (15.0) | 114 (10.3) | 0 | 0 | 0 | 0 |
| | (=) | 0 | 61 (5.5) | 405 (36.7) | 53 (4.8) | 0 | 11 (1.0) |
| | (-) | 0 | 0 | 0 | 106 (9.6) | 189 (17.1) | 0 |

The symbol between brackets indicates the right and left hip. (+: increased; =: no change; -: decreased).

Footnote:

The least significant change (LSC) of the right total hip and left total hip were used to compare the BMD changes when monitoring the total hip BMD of each hip separately.

The LSC of mean bilateral total hip BMD was used to compare the BMD changes when monitoring the mean bilateral total hip BMD.

separately, whereas there was no significant change according to monitoring the mean bilateral total hip BMD. There was a significantly lower BMD change in 4.8% of the participants undergoing monitoring total hip BMD of each hip separately; however, this was not considered significant according to the monitoring the mean bilateral total hip BMD. Additionally, 1.0% of the participants had a significantly higher BMD change involving one hip and a significantly lower BMD change at the contralateral hip, which was not significant when monitoring the mean bilateral total hip BMD. These were considered discordant and the discordant percentage was 11.3% between the monitored mean bilateral total hip BMD and the total hip BMD of each hip separately in follow-up BMD measurements. In other words, 11.3% of the participants had an undetected change when monitoring the mean bilateral total hip BMD, compared with both hips separately. This indicates that analyzing both hips separately is more sensitive for changes than using the mean bilateral total hip BMD. In contrast, the average LSC of the right and left total hip BMD, rather than the LSC of the mean bilateral total hip, was used to compare differences in monitoring the mean bilateral total hip BMD in follow-up BMD measurements; the discordance percentage was 19.5% (data not shown).

A comparison of the different monitoring methods between the spine and mean bilateral total hip BMD was showed in Table 3. There was a significantly higher BMD change in the spinal BMD among 15.6% of participants, whereas there was no significant difference in the mean bilateral total BMD. Additionally, 7.1% of the participants had no BMD change involving the spine; however, they had significantly higher BMD change in the mean bilateral total hip BMD. There was a significantly higher BMD change in the spinal BMD among 3.0% of the participants, whereas there was a significantly lower BMD change in the mean bilateral total hip BMD. Additionally, 0.5% of the participants had significantly lower spinal BMD changes; however, this was considered a significantly higher change in the mean bilateral total hip BMD. In contrast, there was no significant change in spinal BMD in 11.1% of the participants, whereas there was a significantly lower change in the mean bilateral total hip BMD. Additionally, 6.2% of the participants had significantly lower spinal BMD changes, but there was no significant change in the mean bilateral total BMD. The discordance percentage was 43.5% between the

**Table 3. Comparison of changes between monitoring spine BMD and mean bilateral total hip BMD on repeat tests.**

| | | Changes of spine BMD, n (%) | | | |
|---|---|---|---|---|---|
| | | (+) | (=) | (-) | Kappa |
| Changes of mean bilateral total hip BMD, n (%) | (+) | 197 (17.8) | 78 (7.1) | 5 (0.5) | 0.326 |
| | (=) | 172 (15.6) | 290 (26.2) | 68 (6.2) | |
| | (-) | 33 (3.0) | 123 (11.1) | 139 (12.6) | |

+: increased; =: no change; -: decreased.

monitored spine and the mean bilateral total hip BMD in follow-up BMD measurements. Cohen's kappa coefficient was 0.326, indicating poor agreement. Our study was not to directly compare the magnitude of change between these two sites, but rather to point out the clinical challenge of interpreting discordant changes when both are routinely monitored in follow-up assessments.

Table 4 presented a comparison of changes between spinal BMD and TBS on repeat tests. There was a significantly higher change among in the spinal BMD among 27.3% of the participants, whereas there was no significant change in the TBS. Additionally, 4.1% of the participants had no change in the spinal BMD; however, they had significantly greater TBS changes. There was a significantly higher change in the spinal BMD among 4.7% of the participants, whereas there was a significantly lower change in the TBS. Additionally, 1.4% of the participants had significantly lower changes in the spinal BMD; however, they had a significantly higher change in the TBS. In contrast, there was no significant change in the spinal BMD among 6.1% of the participants, whereas there was a significantly lower TBS change. Additionally, 14.4% of the participants had significantly lower changes in the spinal BMD, but no significant change in the TBS. The discordance percentage between the monitored spinal BMD and TBS in follow-up BMD measurements was 58.0%. Cohen's kappa coefficient was 0.034, indicating poor agreement. Our result was not in reference to direct equivalence between BMD and TBS, but rather to describe the concordance or discordance in their longitudinal change patterns during follow-up.

## Discussion

In this study, we monitored changes in BMD based on various follow-up BMD measurements. When using the LSC of the mean bilateral total hip BMD, the discordance between the mean bilateral total hip BMD and the total hip BMD of each hip separately was 11.3%. In contrast, when using the average LSC of the right and left total hip BMD, the discordance increased to 19.5%. This suggests that analyzing each hip separately is more sensitive to detecting changes than using the mean bilateral total hip BMD, and the LSC of the mean bilateral total hip BMD is recommended. Additionally, the discordance between the lumbar spine and the mean bilateral total hip BMD was 43.5%, while the discordance between lumbar spine BMD and TBS reached 58.0%, reflecting the differing biological responses and measurement characteristics of these parameters.

The 2023 updated ISCD Adult Official Positions recommends that the mean bilateral total hip BMD should be used for monitoring when both hips have been scanned in follow-up BMD measurements [2]. However, they did not mention how to determine the LSC, such as the mean LSC of bilateral total hip BMD or the average LSC of the right and left total hip BMD. In our study, the LSC of the mean bilateral total hip BMD was lower than the average LSC of the right and left total hip BMD. When the LSC of the mean bilateral total hip BMD was used to compare the differences in follow-up BMD measurements, the discordant percentage was 11.3%. In contrast, when the average LSC of the right and left total hip BMD was used, the discordant percentage was 19.5%. Therefore, the LSC of the mean bilateral total hip BMD should be used to monitor the mean bilateral total hip BMD during follow-up BMD measurements.

Our results revealed that the discordance percentage was 11.3% between the monitored mean bilateral total hip BMD and the total hip BMD of each hip separately. Among the discordant results, 4.8% of the participants had significantly

Table 4. Comparison of changes between spine BMD and trabecular bone score on repeat tests.

| | | Changes of spine BMD, n (%) | | | |
| --- | --- | --- | --- | --- | --- |
| | | (+) | (=) | (-) | Kappa |
| Changes of trabecular bone score, n (%) | (+) | 48(4.3) | 45(4.1) | 15(1.4) | 0.034 |
| | (=) | 302(27.3) | 379(34.3) | 159(14.4) | |
| | (-) | 52(4.7) | 67(6.1) | 38(3.4) | |

+: increased; =: no change; -: decreased.

decreased BMD at one side of the hip and no significant change in the mean bilateral total hip BMD. Additionally, 1.0% of the participants had significantly increased BMD involving one hip and decreased BMD at the other hip, which was not significant when monitoring the mean bilateral total hip BMD. This suggests that analyzing each hip separately is more sensitive to detecting changes than using the mean bilateral total hip BMD. Relying solely on the mean may result in overlooking patients who require more aggressive management. While monitoring the mean bilateral total hip BMD is a simpler and more convenient approach for tracking changes in both hips, it can potentially mask significant changes occurring in one hip. Although this method is appropriate for most patients, in clinical situations where greater sensitivity to localized changes is needed—such as during early treatment response or in high-risk individuals—it may be more appropriate to monitor each hip separately. These findings underscore the importance of adopting tailored monitoring strategies based on the clinical context.

Chen et al. reported that a diagnosis based on the use of the lumbar spine and one site of hip BMD failed to identify 13.4% to 15.8% of osteoporotic subjects, which may indicate that simultaneously monitoring the spine and both hips represented the best practice for DXA BMD measurements [6]. There was spine-hip discordance between the monitored spine and mean bilateral total hip BMD in follow-up BMD measurements in our study and the discordance percentage was 43.5%. Kolta et al. also reported a significant decrease in the hip BMD, whereas the spinal BMD did not change during a 6-year follow-up [7]. The spine was more affected by age-related degenerative changes than the hip. Liu et al. reported that lumbar osteophytes explained 16.6% of measured changes in the spinal BMD in women [8]. In a study of osteoporotic fractures, women with degenerative lumbar disease showed a 9% to 13% increase in the spinal BMD [9]. Ichchou et al. demonstrated that degenerative lumbar disease could increase spinal BMD by up to 15% in women with severe lumbar osteoarthritis [10]. The causes of the spine-hip discordance may be as follows. First, the peak bone mass and rate of bone loss are not the same for different skeletons [11]. Second, the initial rate of bone loss was greater in the trabecular bone than in the cortical bone in postmenopausal women [12]. Third, spine was more affected by age-related degenerative changes than the hip [8]. Therefore, a significant increase in spinal BMD on repeat testing may result from either true bone gain or degenerative changes. If total hip BMD shows a significant decrease during follow-up, aggressive management may still be warranted despite an apparent increase in spinal BMD. Conversely, a significant decrease in spinal BMD likely indicates true bone loss. In such cases, aggressive management may still be necessary even if total hip BMD has increased. Additionally, rather than implying equivalence or allowing direct comparisons between these two skeletal sites, our study highlights the importance of structured interpretation frameworks to help clinicians make informed decisions when faced with discordant BMD findings.

BMD and TBS are fundamentally different but complementary measures—reflecting bone quantity and bone microarchitecture, respectively. According to the literature, increased soft tissue over the lumbar spine can affect both spinal BMD and TBS measurements. While BMD may be influenced, the change typically does not exceed the LSC, and therefore is unlikely to have clinical significance. In contrast, greater soft tissue thickness is associated with a lower TBS value [13,14]. However, it is believed that TBS is not affected by lumbar degenerative change compared to spine BMD. Dufour et al. reported that osteoarthritis at L4 caused a 19% increase in spine BMD at L4; however, there was no significant change in the TBS [15]. Kolta et al. also demonstrated that lumbar osteoarthritis caused an increase in spinal BMD in patients with lumbar osteoarthritis, whereas TBS was not affected [7]. Additionally, during the follow-up period, there was a significant decrease in the TBS, whereas spinal BMD was unchanged [7]. In our study, there was also a discordance between spinal BMD and TBS in follow-up BMD measurements, with a discordance percentage of 58.0%. From a clinical perspective, discordance between BMD and TBS—such as an improvement in BMD accompanied by a decline in TBS (or vice versa)—can complicate the interpretation of treatment response or changes in fracture risk. Although the ISCD recommends monitoring spinal BMD over TBS [2], the precise role of TBS in tracking the progression of treated or untreated osteoporosis remains uncertain. Generally, anabolic agents produce more pronounced changes in TBS compared to anti-resorptive therapies [5]. Further research is needed to clarify the utility of TBS in monitoring therapeutic response.

The strength of this study lies in the inclusion of a substantial number of participants who underwent follow-up BMD measurements of the spine and both hips. Second, to the best of our knowledge, this is the first study to compare different methods for monitoring BMD changes. Nevertheless, this study had some limitations. First, it was conducted in postmeno-pausal Asian women, potentially affecting the generalizability of our results. Second, BMD and TBS were measured using Hologic instruments only. However, further studies are required to confirm these findings.

In conclusion, our study demonstrated that analyzing each hip separately is more sensitive for detecting changes than using the mean bilateral total hip BMD. When monitoring the mean bilateral total hip BMD on repeat tests, it is recommended to use the LSC of the mean bilateral total hip BMD. Discordance between spine and hip BMD, as well as between spinal BMD and TBS, was observed in follow-up measurements. These findings emphasize the need for structured inter-pretation frameworks to guide clinical decision-making in the presence of discordant results.

## Acknowledgments

The requirement of informed consent was waived due to the retrospective nature of this study.

## Author contributions

**Conceptualization:** Yen-Huai Lin.

**Data curation:** Yen-Huai Lin.

**Formal analysis:** Yen-Huai Lin.

**Funding acquisition:** Yen-Huai Lin.

**Investigation:** Yen-Huai Lin.

**Methodology:** Yen-Huai Lin.

**Project administration:** Yen-Huai Lin.

**Resources:** Yen-Huai Lin, Michael Mu Huo Teng.

**Software:** Yen-Huai Lin.

**Supervision:** Yen-Huai Lin, Michael Mu Huo Teng.

**Validation:** Yen-Huai Lin.

**Visualization:** Yen-Huai Lin, Michael Mu Huo Teng.

**Writing – original draft:** Yen-Huai Lin.

**Writing – review & editing:** Yen-Huai Lin, Michael Mu Huo Teng.

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
