## [Decision Letter · Decision Letter 0]

11 Jun 2025

Dear Dr. Teng,

Thank you for submitting your manuscript to PLOS ONE. After careful consideration, we feel that it has merit but does not fully meet PLOS ONE’s publication criteria as it currently stands. Therefore, we invite you to submit a revised version of the manuscript that addresses the points raised during the review process.

We look forward to receiving your revised manuscript.

Kind regards,

Gaetano Paride Arcidiacono

Academic Editor

PLOS ONE

Journal Requirements:

“This work was supported by a grant (CHGH114-113-N23) from the Cheng Hsin General Hospital, Taipei, Taiwan.”

Reviewers' comments:

Reviewer's Responses to Questions

**Comments to the Author**

1. Is the manuscript technically sound, and do the data support the conclusions?

Reviewer #1: Partly

2. Has the statistical analysis been performed appropriately and rigorously?

Reviewer #1: Yes

3. Have the authors made all data underlying the findings in their manuscript fully available?

Reviewer #1: No

4. Is the manuscript presented in an intelligible fashion and written in standard English?

Reviewer #1: Yes

Reviewer #1: This manuscript reports on methods to assess and evaluate changes in DXA derived bone outcomes over time. In particular, differences in hip BMD when the mean of both hips is analyzed vs both hips separately, differences in change between hip and LS BMD, and differences in change between LS BMD and TBS. Generally, the study seems well conducted, although I do not fully agree on the interpretation of the data.

Generally:

Sometimes the story is difficult to follow, especially the hip comparison. The authors use: mean bilateral total hip BMD vs. total hip BMD of both hips. I would recommend using: mean bilateral total hip BMD vs. total hip BMD of each hip separately.

Introduction:

“However, the ISCD does not mention how to interpret BMD changes when both the regions are scanned during follow-up BMD measurements. Therefore, discordant results may be obtained when simultaneously monitoring BMD changes in the spine and hips.”

Is this really a problem? There are many factors involved in BMD changes over time. For example, different (un)loading patterns of the spine and hip may strongly influence the magnitude and direction of change for both sites.

General for entire introduction: Yes there can be discordance in the magnitude or direction of change between BMD of different scans or different sites. But I am not sure whether this is really important. In many case, the discordance can be explained by specific loading patterns or interventions that can affect some sites or outcomes more than the other. This also justifies the rationale to conduct multiple scans. The interpretation of the results can than be based on multiple scans….

The theoretical framework would be strengthened by a more comprehensive integration of the relevant literature.

Methods:

Cohort: What was the median or mean follow-up time?

BMD changes have been reduced to an ordinal scale. Have the authors also considered to address the agreement between changes on a continuous scale? In this case, also the magnitude and direction of changes between outcomes can be assessed.

Results:

Table 1: Can you also provide T and Z-scores for sites? This provides an indication how participants’ BMD can be interpreted relative to general population. Was it a cohort characterized by osteoporosis?

Table 2: Table legend needs to be improved. I assume that the symbol between brackets indicate the right and left hip? This is nowhere explicitly mentioned.

Table 2: Alternative explanation: When using the mean of both hips, 5.5% shows no change, while a decrease would have been detected if both hips were analyzed separately. When using the mean of both hips, 4.8% shows no change, while an increase would have been detected if both hips were analyzed separately. In other words, 10.3% (+1.0%) of the participants has and undetected change when the mean of both hips is analyzed, compared with both hips separately. Imo, this shows that analyzing both hips separately is more sensitive for changes than using the mean. But this cannot be regarded as poor agreement.

Table 3: Imo, it does not really does not really make sense to compare the hip with LS. Both sites respond differently to similar loading stimuli and interventions. The outcomes should also be interpreted as such. Anyhow, it does not mean poor agreement.

Table 4: BMD and TBS are two different outcomes. Imo, the Table indicates that TBS is less sensitive to changes over time than LS. But that does not mean poor agreement per se. It could indicate, however, that TBS is less suited to evaluate or monitor BMD status.

Conclusion:

“Additionally, the discordance percentage was 43.5% between the monitored spine and the mean bilateral total hip BMD, which indicated poor agreement.”

This statement is not justified. You would never aspect good agreement between both sites.

“We also compared changes between the monitored spinal BMD and TBS in follow-up BMD measurements; the discordance percentage was 58.0%, which also showed poor agreement”

This statement is also not justified. When one measurement is not sensitive to changes over time, while the other is relatively sensitive, I would never expect good agreement.

“When the LSC of the mean bilateral total hip BMD was used to compare the differences in follow-up BMD measurements, the discordant percentage was 11.3%.”

Is this really discordance? When taking a mean of two measurements, it just becomes less sensitive. Thus, a true change in of two hips, may go undetected when taking the mean of two hips. My interpretation is that when high sensitivity is required, it could be more beneficial to monitor and interpret both hips separately.

The discussion would be strengthened by a more comprehensive integration of the relevant literature.

**Do you want your identity to be public for this peer review?** For information about this choice, including consent withdrawal, please see our Privacy Policy

Reviewer #1: No

---

## [Author Response · Author response to Decision Letter 1]

28 Jul 2025

1. Is the manuscript technically sound, and do the data support the conclusions?

Reviewer #1: Partly

Response: Thank you for your comment. Our aim is to emphasize the need for structured interpretive frameworks that help clinicians understand and respond to discordant findings, rather than to suggest equivalence or direct comparisons between measurement sites or between BMD and TBS. To clarify this distinction, we have revised the manuscript to avoid implying that such discordance is inherently negative. Instead, we highlight that these findings underscore the importance of site- and modality-specific interpretation in the longitudinal monitoring of bone health. This issue is especially relevant in routine follow-ups, where treatment decisions often hinge on the perceived response to therapy. We believe that recognizing this interpretive ambiguity further supports the need for clearer clinical guidance. In response, we have clarified our discussion, incorporated additional relevant literature, and conducted a further round of English language editing to improve clarity and readability. We hope these revisions enhance the scientific contribution of our work.

2. Has the statistical analysis been performed appropriately and rigorously?

Reviewer #1: Yes

Response: Thank you for your comment.

3. Have the authors made all data underlying the findings in their manuscript fully available?

Reviewer #1: No

Response: Thank you for your comment. The data set is owned by the institutional review board of Cheng Hsin General Hospital. The institutional review board of Cheng Hsin General Hospital only approved the data analysis in our study and did not approve data sharing. Therefore, we do not have permission to share the data set. Interested researchers can submit data access requests to the institutional review board of Cheng Hsin General Hospital through the following email address: chghirb@chgh.org.tw Others would be able to access the data in the same manner as the authors. We have included the data availability statement on page 18, line 321.

Page 18, line 316

Data Availability statement:

The dataset used in this study is owned by the Institutional Review Board (IRB) of Cheng Hsin General Hospital. While the IRB approved our use of the data for analysis, it did not grant permission for data sharing. As a result, we are unable to make the dataset publicly available. However, interested researchers may request access by contacting the IRB of Cheng Hsin General Hospital at chghirb@chgh.org.tw. Access will be provided under the same conditions as those granted to the authors, who do not have any special privileges.

4. Is the manuscript presented in an intelligible fashion and written in standard English?

Reviewer #1: Yes

Response: Thank you for your comment.

5. Review Comments to the Author

Reviewer #1: This manuscript reports on methods to assess and evaluate changes in DXA derived bone outcomes over time. In particular, differences in hip BMD when the mean of both hips is analyzed vs both hips separately, differences in change between hip and LS BMD, and differences in change between LS BMD and TBS. Generally, the study seems well conducted, although I do not fully agree on the interpretation of the data.

1. Generally:

Sometimes the story is difficult to follow, especially the hip comparison. The authors use: mean bilateral total hip BMD vs. total hip BMD of both hips. I would recommend using: mean bilateral total hip BMD vs. total hip BMD of each hip separately.

Response: Thank you for your comment. We have revised the manuscript on page 2, line 20, page 4, line 66, and page 5, line 86.

Page 2, line 20

Monitoring bone mineral density (BMD) changes can be challenging when both the spine and hips are included in follow-up assessments. This study aimed to evaluate BMD changes using different monitoring approaches, including comparisons between the mean bilateral total hip BMD and total hip BMD of each hip separately, spinal BMD and mean bilateral total hip BMD, and spinal BMD and trabecular bone score (TBS).

Page 4, line 66

The International Society for Clinical Densitometry (ISCD) recommends that when both hips are scanned on repeat tests, the mean bilateral total hip BMD should be used for monitoring [2]. However, the differences between the monitored mean bilateral total hip BMD and the total hip BMD of each hip separately are uncertain.

Page 5, line 86

Therefore, the aim of this study was to compare different monitoring methods (mean bilateral total hip BMD vs. total hip BMD of each hip separately, spinal BMD vs. mean bilateral total hip BMD, spinal BMD vs. TBS); additionally, we also aimed to determine LSC when monitoring mean bilateral total hip BMD in follow-up BMD measurements.

2. Introduction:

“However, the ISCD does not mention how to interpret BMD changes when both the regions are scanned during follow-up BMD measurements. Therefore, discordant results may be obtained when simultaneously monitoring BMD changes in the spine and hips.”

Is this really a problem? There are many factors involved in BMD changes over time. For example, different (un)loading patterns of the spine and hip may strongly influence the magnitude and direction of change for both sites.

General for entire introduction: Yes there can be discordance in the magnitude or direction of change between BMD of different scans or different sites. But I am not sure whether this is really important. In many case, the discordance can be explained by specific loading patterns or interventions that can affect some sites or outcomes more than the other. This also justifies the rationale to conduct multiple scans. The interpretation of the results can than be based on multiple scans….

The theoretical framework would be strengthened by a more comprehensive integration of the relevant literature.

Response: We appreciate the reviewer’s thoughtful observation regarding the discordance in BMD changes between different skeletal sites, such as the spine and both hips. Indeed, as correctly noted, there are several factors—including mechanical loading patterns, pharmacological interventions, and degenerative changes—that can contribute to differential changes in BMD at these sites over time. This variability is one of the reasons why multiple-site assessment is recommended in clinical practice.

However, our point was not to question the need for multiple-site measurements, but rather to highlight a gap in current clinical guidelines. Specifically, the ISCD does not provide explicit guidance on how to interpret discordant longitudinal BMD changes when the spine and both hips are assessed during follow-up. While individual site changes can often be explained retrospectively, there is limited standardization or consensus on how to weigh such differences when making clinical decisions—particularly in cases where one site shows improvement while another shows deterioration.

This lack of standardized interpretation can be especially relevant in clinical trials or routine follow-ups where treatment decisions (e.g., initiation or discontinuation of osteoporosis therapy) hinge on the perceived response. As such, we believe that acknowledging this ambiguity reinforces the need for clearer interpretive frameworks in longitudinal BMD monitoring and justifies our focus on this issue. We have revised the manuscript on page 4, line 73.

Page 4, line 73

Various factors such as mechanical loading, pharmacological interventions, and degenerative changes can lead to site-specific variations in BMD over time, supporting the use of multiple-site assessments in clinical practice. The ISCD recommends measuring BMD at both the spine and hip for all patients [2]. However, the ISCD does not provide clear recommendations for interpreting discordant BMD changes when the spine and hips are assessed during follow-up. While such differences can sometimes be explained retrospectively, there is no standardized approach for incorporating them into clinical decision-making, especially when one site improves and another worsens. This ambiguity is particularly relevant in settings where treatment decisions rely on BMD trends.

The ISCD also recommends evaluating the bone quality using the trabecular bone score (TBS) [4]. When monitoring changes in the spinal BMD and TBS simultaneously on repeat tests, practitioners are often uncertain about the association between monitoring spinal BMD and TBS. Therefore, the aim of this study was to compare different monitoring methods (mean bilateral total hip BMD vs. total hip BMD of each hip separately, spinal BMD vs. mean bilateral total hip BMD, spinal BMD vs. TBS); additionally, we also aimed to determine LSC when monitoring mean bilateral total hip BMD in follow-up BMD measurements. Our aim was to highlight a gap in current guidelines, and our focus on this issue emphasizes the need for clearer guidance in interpreting longitudinal BMD data.

3. Methods:

Cohort: What was the median or mean follow-up time?

Response: Thank you for your comment. We have revised the manuscript on page 6, line 101.

Page 6, line 101

Finally, 1,105 participants were enrolled in the study and the mean follow-up time was 23 months.

4. BMD changes have been reduced to an ordinal scale. Have the authors also considered to address the agreement between changes on a continuous scale? In this case, also the magnitude and direction of changes between outcomes can be assessed.

Response: We thank the reviewer for this insightful comment. In our study, we primarily categorized BMD changes using an ordinal scale (e.g., decreased, no change, increased) to support clinical interpretation and to align with commonly used thresholds in practice, such as the least significant change (LSC). The direction of change is also effectively captured using this approach. While continuous scales can provide additional detail, averaging BMD values—such as from both hips—can have a statistical dampening effect, potentially attenuating the apparent magnitude of changes and reducing sensitivity compared to site-specific assessments. Additionally, our intention was not to directly compare the absolute magnitude of change between sites, but rather to point out the clinical challenge of interpreting discordant changes when both are routinely monitored in follow-up assessments. For these reasons, we focused our primary analysis on the ordinal scale, given its clinical relevance and the limitations associated with averaging on a continuous scale.

5. Results:

Table 1: Can you also provide T and Z-scores for sites? This provides an indication how participants’ BMD can be interpreted relative to general population. Was it a cohort characterized by osteoporosis?

Response: Thank you for your comment. This study included only postmenopausal women, for whom the International Society for Clinical Densitometry (ISCD) recommends using T-scores rather than Z-scores. Accordingly, we have reported T-scores for each site and revised Table 1 to reflect this. The observed prevalence of osteoporosis was 70.1%, which is likely attributable to the hospital-based nature of the study population.

6. Table 2: Table legend needs to be improved. I assume that the symbol between brackets indicate the right and left hip? This is nowhere explicitly mentioned.

Response: Thank you for your comment. Accordingly, we have revised Table 2.

7. Table 2: Alternative explanation: When using the mean of both hips, 5.5% shows no change, while a decrease would have been detected if both hips were analyzed separately. When using the mean of both hips, 4.8% shows no change, while an increase would have been detected if both hips were analyzed separately. In other words, 10.3% (+1.0%) of the participants has and undetected change when the mean of both hips is analyzed, compared with both hips separately. Imo, this shows that analyzing both hips separately is more sensitive for changes than using the mean. But this cannot be regarded as poor agreement.

Response: Thank you for your comment. We have revised the manuscript on page 9, line 156 and page 14, line 231.

Page 9, line 156

In other words, 11.3% of the participants had an undetected change when monitoring the mean bilateral total hip BMD, compared with both hips separately. This indicates that analyzing both hips separately is more sensitive for changes than using the mean bilateral total hip BMD.

Page 14, line 231

Our results revealed that the discordance percentage was 11.3% between the monitored mean bilateral total hip BMD and the individual total hip BMDs (right and left hips). Among the discordant results, 4.8% of the participants had significantly decreased BMD at one side of the hip and no significant change in the mean bilateral total hip BMD. Additionally, 1.0% of the participants had significantly increased BMD involving one hip and decreased BMD at the other hip, which was not significant when monitoring the mean bilateral total hip BMD. This suggests that analyzing each hip separately is more sensitive to detecting changes than using the mean bilateral total hip BMD. Relying solely on the mean may result in overlooking patients who require more aggressive management. While monitoring the mean bilateral total hip BMD is a simpler and more convenient approach for tracking changes in both hips, it can potentially mask significant changes occurring in one hip. Although this method is appropriate for most patients, in clinical situations where greater sensitivity to localized changes is needed—such as during early treatment response or in high-risk individuals—it may be more appropriate to monitor each hip separately. These findings underscore the importance of adopting tailored monitoring strategies based on the clinical context.

8. Table 3: Imo, it does not really does not really make sense to compare the hip with LS. Both sites respond differently to similar loading stimuli and interventions. The outcomes should also be interpreted as such. Anyhow, it does not mean poor agreement.

Response: Thank you for your comment. In our study, we used the term “agreement” to describe the concordance or discordance in their longitudinal change patterns during follow-up. We are specifically reporting the statistical measure of Cohen’s kappa. We agree that the lumbar spine and hip respond differently to mechanical loading and therapeutic interventions, and that these differences should be acknowledged when interpreting BMD changes. However, our intention was not to directly compare the magnitude of change between the two sites, but rather to point out the clinical challenge of interpreting discordant changes when both are routinely monitored in follow-up assessments.

In practice, both the spine and hip are measured preci

---

## [Decision Letter · Decision Letter 1]

7 Dec 2025

Comparison of different methods for monitoring changes in bone mineral density during follow-up measurements

PONE-D-25-13444R1

Dear Dr. Teng,

We’re pleased to inform you that your manuscript has been judged scientifically suitable for publication and will be formally accepted for publication once it meets all outstanding technical requirements.

Kind regards,

Gaetano Paride Arcidiacono

Academic Editor

PLOS One

---

## [Editor Report · Acceptance letter]

PONE-D-25-13444R1

PLOS One

Dear Dr. Teng,

I'm pleased to inform you that your manuscript has been deemed suitable for publication in PLOS One. Congratulations! Your manuscript is now being handed over to our production team.

Kind regards,

on behalf of

Dr. Gaetano Paride Arcidiacono

Academic Editor

PLOS One